# Peeling Force Required for the Detachment of Non-Woven Plastic Tissue from the Surface of Mortar Prisms

**DOI:** 10.3390/polym15214286

**Published:** 2023-10-31

**Authors:** Sifatullah Bahij, Safiullah Omary, Essia Belhaj, Vincent Steiner, Francoise Feugeas

**Affiliations:** 1ICube UMR 7357, Laboratoire des Sciences de l’Ingénieur, de l’Informatique et de l’Imagerie, INSA de Strasbourg, University of Strasbourg, 24 Boulevard de la Victoire, 67084 Strasbourg, France; sifatullah.bahij@insa-strasbourg.fr (S.B.);; 2Department of Civil and Industrial Construction, Kabul Polytechnic University, Kabul 1001, Afghanistan

**Keywords:** cement mortar, non-woven sheets, adhesion properties, peeling force, interferometry analysis, microscopic visualization, mechanical strengths

## Abstract

The purpose of this experimental paper is to examine the adhesion properties between non-woven plastic sheets and cement mortar. Specifically, the effect of w/c ratio and quantity of superplasticizer on the peeling force required for the detachment of tissue from the surface of prisms was studied in detail. Therefore, two types of mortar mixtures were prepared: (1) mixtures without superplasticizer with three different w/c ratios of 0.45, 0.50, and 0.55, and (2) mixtures with reduced amounts of water and three various percentages of superplasticizer of 0.0%, 1.11%, and 2.17% (by weight of cement). For this purpose, bond tests with a special setup, interferometry and microscopic analyses, and mechanical tests were performed. The results highlight that non-woven sheets had strong adhesion to cement mortar without using any adhesive materials. However, the peeling force improved by 15.78% as the w/c ratio increased from 0.50 to 0.55. Conversely, this force declined by 24.50% as the w/c ratio decreased from 0.50 to 0.45. In addition, the peeling force decreased by 20.62% as the w/c ratio decreased from 0.50 to 0.45 and 1.11% superplasticizer was added to the mixtures. This property decreased further by 38.29% as the w/c ratio lowered to 0.40, and the amount of superplasticizer increased to 2.17%. The interferometry and microscopic analyses clearly demonstrate that the adhesion between tissue and mortar is largely related to the surface texture, amount of cement paste, and quantity of residual fibers on the surfaces of samples. It indicates that mortar samples with higher w/c ratios had a smoother surface, and providing more contact area for microfilaments, which resulted in thicker layers of remaining fibers compared to the specimens with a lower w/c ratio. Even though there was not much difference in the surface texture of specimens with superplasticizer and lower w/c ratios, because of their similar workability. Still, thicker layers of microfilaments remained on the surface of specimens containing a lower amount of superplasticizer, which resulted in strong adhesion between sheet and cement mortar.

## 1. Introduction

Concrete is widely employed in the construction sector owing to its economical nature and capacity to be molded into various dimensions and forms. Concrete structures may deteriorate with time for many reasons, such as aging, faults in design, change in usage, inadequate maintenance, material faults, excess loads, environmental facets, etc. [1]. Therefore, the deteriorated structures could be replaced by new ones or can be repaired. Here, the replacement could be the costly option, which opens the need for the strengthening of structural elements [1,2,3].

The strengthening or repair of concrete structures is not a new topic, but plenty of research has been performed to study the efficiency of strengthened or repaired concrete elements [4,5,6,7]. For this purpose, different types of cementitious materials, such as ultra-high performance concrete (UHPC) [8,9], reinforced mortar layer [10], cementitious grout [11], high-performance fiber reinforced concrete (HPFRC) [12], engineered cementitious composites (ECCs), and ultra-high performance fiber-reinforced cementitious composites (UHPFRCCs) [13] were used. In addition, various fiber-reinforced polymer (FRP) composites, such as carbon fiber-reinforced polymer (CFRP) [14], glass fiber-reinforced polymer (GFRP) [15], basalt fiber-reinforced polymer (BFRP) [16], and aramid fiber reinforced polymer (AFRP) [17], were applied to strengthen concrete elements. In addition, such materials have the potential to strengthen various structural components, such as beams, columns, beam-column joints, masonry walls, etc., to effectively enhance their resistance to flexure, shear, torsion, seismic, impact, or other loading conditions [18,19,20].

For instance, a research study was performed to investigate the flexural behavior of reinforced concrete (RC) beams strengthened by CFRP strips using the side near surface mounted (SNSM) technique and epoxy as adhesive materials. The findings indicate that the ultimate load capacity of the strengthened beams was improved significantly compared to the control ones. This improvement was more significant for the beams strengthened vertically compared to the horizontal ones. Moreover, the strengthened beams had narrower crack widths compared to the unstrengthened ones. In addition, the crack widths were narrower in vertically strengthened beams compared to the horizontal ones [2].

Likewise, an experimental investigation was carried out to assess the shear characteristics of RC beams that were reinforced with jackets made of high-performance fiber-reinforced concrete (HPFRC). Overall, the shear load capacity and mid-span deflection were enhanced for the strengthened beams when compared to the unstrengthen ones. Here, the beams with thixotropic material on their lateral sides exhibited lower shear load capacity and deflection in comparison to those with self-leveling material. Additionally, the retrofitted beams exhibited certain ductile behaviors and improved stiffness as compared to the control ones [21].

Similarly, research work was carried out to examine the torsional behaviors of RC beams strengthened with near surface mounted-fiber reinforced polymer (NSM-FRP) using epoxy or cement-based adhesives. The outputs highlight that the torsional load capacity and twisting angle were enhanced considerably for the strengthened beams. Moreover, this enhancement was more significant for the beams using epoxy as an adhesive material compared to the cement-based mortar [22].

Recently, a novel category of materials called non-woven sheets has been employed to enhance specific properties of mortar and concrete without the use of a bonding agent. These sheets were used in different configurations, such as 1-layer, 2-faces, 3-faces, and complete wrapping to strengthen mortar/concrete specimens. The results indicate a significant improvement in the compressive, flexural, and split tensile strengths of the retrofitted samples compared to the reference ones. Furthermore, this type of tissue demonstrated notable advantages in enhancing the cracking mechanism of the reinforced samples. In contrast, the control specimens experienced damage to multiple parts in the case of compressive strength and separation into two separate parts during flexural and split tensile tests. Conversely, the samples reinforced with non-woven tissue remained together, even reaching the ultimate loads, especially in the case of 3-faces or complete wrapping [23,24,25].

It was observed from the literature that researchers used various adhesive materials to bond the strengthening materials to the concrete components. For example, the most applied materials were epoxy, sandblasting, cement-based mortar, etc. [22]. In addition, the authors found that the structural behaviors of strengthened members were greatly related to the bond behaviors between strengthening materials and original the structural members.

As previously mentioned, the utilization of non-woven fabrics had the capability to enhance certain properties of cement-based materials. Since adhesive materials were not employed, the fact behind the adhesion between tissue and cementitious materials and the affecting factors are still unknown and could be counted as an important subject for research. Therefore, in this research work, non-woven sheets manufactured by the Freudenberg company located in Weinheim, Germany [26] were used to strengthen the specimens.

Non-woven fabrics are broadly characterized as sheet-like or web-like structures formed by combining short and long fibers, which are then bonded through chemical, mechanical, heat, or solvent treatments [27,28,29]. These sheets have a flat, porous nature and have isotropic or non-isotropic behaviors. The production process of such sheets is relatively short, resulting in their economic advantage over knitted or woven fabrics. The materials used for the manufacture of such fabrics could be natural, synthetic, or semi-synthetic, such as polylactic acid (PLA), polypropylene (PP), polyethersulfone (PES), polyethylene (PE), polyethylene terephthalate (PET), polyamide (PA), etc. In addition, non-woven tissue has many applications, especially in disposable or single-use products in schools, hospitals, nursing homes, and luxury accommodations [26].

In order to evaluate the adhesion behaviors between non-woven sheets and cement mortar, three various w/c ratios of 0.45, 0.50, and 0.55, and three different percentages of SP content (0%, 1.11%, and 2.17%) were selected as affecting parameters. Firstly, the 180° pull-out bond test [30] was performed with the help of a special setup prepared inside the laboratory of INSA de Strasbourg to measure the peeling force required for the removal of the fabric from the prisms. Then, interferometry and microscopic analyses were carried out to investigate the facts behind the bond behaviors and the influence of each affecting parameter. Finally, the flexural and compressive strengths were measured on the same samples after the bond test to verify the adhesion of the remaining microfilaments within the mortar mixtures.

## 2. Experimental Program

### 2.1. Materials

Mortar mixtures were prepared using an Ordinary Portland Cement of CEM I 52.5 N confirming NF EN 197-1/A1 standard, and manufactured by EQIOM located in Heming, France [31]. This cement had a specific gravity of 3.11 g/cm^3^ and a surface area determined through laser granulometry of 5600 cm^2^/g. The increase in compressive strength over time and the chemical composition of the cement, as detailed in the technical documentation provided by the manufacturer, are illustrated in Figure 1 and Table 1.

In addition, the fine aggregates were obtained from natural river sand, confirming the NF EN 12620/IN1 [32] standard, and had particle sizes of 0–4 mm. The sand had a specific gravity of 2.589 g/cm^3^, a bulk density of 1734.7 kg/m^3^, and a water absorption of 1.076%. The size distribution of such aggregates is shown in Figure 2.

Furthermore, the superplasticizer was sourced from a local supplier, SIKA, in France and certified by EN 934-2 [33]. Table 2 shows the chemical and physical properties of the mentioned superplasticizer.

In the current research work, evolon^®^ non-woven textile [26] manufactured by the Freudenberg company was utilized for all mortar samples, as shown in Figure 3. These fabrics have 70% PET and 30% PA [35] and were manufactured through a process involving splitting, entangling, and bonding using water jets at elevated pressure [36,37,38]. The outcome is a distinctive technical textile characterized by a microfilament arrangement, which has mechanical strength similar to woven textiles along with a soft texture. The physical and mechanical properties of the evolon^®^ non-woven textile are detailed in Table 3.

### 2.2. Preparation of the Specimens

The mortar specimens were cast using a standard mix proportion with the cement and sand ratio of 1:3 [45], as indicated in Table 4.

In order to prepare mortar mixtures with the desired workability, it is essential to take into account proper batching, mixing order, and mixing length. The mixing was performed using a mixer programmed with the standard mixing procedure as follows: (1) Cement and water were added to the mixer and mixed together for a duration of 30 s at a low speed. (2) Subsequently, sand was uniformly added to the mixer for a period of 30 s and mixed for an additional 30 s at a slow speed, followed by 30 s at a higher speed. (3) The mixer was paused for 60 s, during which any mortar adhering to the bottom and sides of the mold was detached and then mixed with the mortar inside the mold. (4) Finally, the mixture was mixed for 60 s at a higher speed [45].

In order to find the adhesion properties between mortar and non-woven sheets, 40 × 40 × 160 mm prisms were prepared. Here, the sheets were cut to a width of 40 mm, corresponding to the prism’s width, and a length of 400 mm (160 mm prism’s length + 240 mm for pulling). First, all sides of the molds were oiled to facilitate easy demolding, and then the cut sheets were positioned within the mold, as shown in Figure 4. Subsequent to completing the mixing procedures, the mortar was poured inside the molds, and the molds were fixed in an automated jolting unit. The entire setup, including the table, mold, hopper, and clamping mechanism, was raised and then dropped a total of 60 times.

Furthermore, 12 prisms were manufactured for each variable parameter. Subsequently, 6 of these samples were tested at 14 days, and the other 6 specimens were tested after 28 days of the curing period. Therefore, the total number of prisms for the 5 variable parameters is equal to 60. The samples inside the molds were placed in a curing room for a duration of 24 h, maintaining curing conditions at a temperature of 23 °C and a humidity of 95%. Following this, the samples were demolded and then cured inside a water curing tank, maintaining the same curing conditions for a maximum duration of 28 days.

The main objective of this article is to investigate the influence of the w/c ratio and the quantity of superplasticizer on the adhesion properties between non-woven tissue and mortar. Therefore, the mixture compositions were formulated by modifying the water content and percentage of superplasticizer, as outlined in Table 5.

In the initial column of the above table, the numerical value signifies the water-to-cement ratio, while the alphabetical value represents whether the mixture includes or excludes the superplasticizer. For example, “0.55-WSP” signifies prisms with a w/c ratio of 0.55 and without a superplasticizer.

### 2.3. Testing Setup and Procedure

To observe the bonding properties, 180° pull-out tests were performed in accordance with the guidelines of ASTM D 903-98 [46]. For such a testing procedure, a metal container was fabricated on-site within the laboratory of INSA de Strasbourg. This container was designed to provide enough space for the proper insertion or removal of the prisms. Furthermore, the container featured a blade at one end to secure the entire setup to the testing machine. While the opposite end was free, the sheet was bent from the bottom of the samples to the top, and the sheet was fixed on the top of the machine, as shown in Figure 5.

The 180° pull-out test was performed with the help of a Shimadzu 100 KN traction machine. The metal container’s blade was secured to the lower jaw, while the unattached end of the fabric sheet was clamped in the upper jaw, as displayed in Figure 6. Once the metal container was firmly in place, the test started, and the machine systematically drew the sheets away from the prisms. During this process, the non-woven sheet detached from the prism’s surface, and measurements of displacement (A) and peeling force (F) were taken at regular intervals. The traction speed was set to 10 mm/min; readings were taken at intervals of Δt = 0.01 s, accumulating an average of 80,000 data points for each individual specimen. Therefore, the test for one sample was completed in (16–20 min) intervals.

In addition, a Bruker Contour GT-K1 3D optical microscope, also known as white and green light interferometry with 2D and 3D measurements, was used to find the surface texture of samples and the concentration of residual microfilaments on the surface of prisms.

After the bond test, samples were investigated under a binocular electronic microscope to provide a more detailed assessment of surface textures, the concentration of microfilaments, and the thickness of remaining fibers on the prisms’ surfaces.

Finally, the flexure and compressive tests were carried out on the same prisms in accordance with EN 1015-11, 2019 [47] code considerations to validate the adhesion of the remaining microfilaments within the mortar mixtures.

## 3. Results and Discussions

### 3.1. Effect of Water Content (w/c Ratio)

As previously mentioned, the mortar mixtures with three distinct w/c ratios were subjected to a 180° pull-out test. The focus was on recording the maximum peeling force required for the detachment of the non-woven tissue from the prisms. The outcomes, as presented in Figure 7, clearly highlight an enhancement in the maximum peeling force as the w/c ratio was raised. Notably, after 28 days of water curing, the peeling force enhanced by 15.78% when the w/c ratio increased from 0.50 to 0.55. In contrast, this force diminished by 24.50% as the water-to-cement ratio reduced from 0.50 to 0.45.

The variation in the peeling force is related to the different surface textures of the samples. It indicates that prisms with higher water content (w/c = 0.55) exhibited more dense microstructures and smoother surfaces with reduced pores that increase the number of points for the attachment of the microfilaments on the prism surfaces. In addition, a sufficient amount of cement paste existed at the surface of prisms having higher water content, and this paste was extensively absorbed by non-woven fabrics; this resulted in higher density, thicker layers, and longer microfilaments remaining on the mortar specimens, as shown in Figure 8a. On the other hand, prisms containing a w/c ratio of 0.45 had rougher surfaces and many irregularities because of an insufficient amount of cement paste, a large number of pores, and fewer adhesion points on the surface. Here, fewer, thinner layers and shorter microfilaments remained on the surfaces of prisms, as shown in Figure 8c. It can be noted that the strong adhesion between non-woven sheets and mortar is due to the presence of high-density, thicker, and longer residual microfilaments.

Furthermore, the specimens were analyzed at a closer view to capture the density of residual fibers, mortar composition, and the locations of fibers that were detached during the bond test. The image clearly shows the intersection and adhesion of microfilaments from non-woven sheets within the mortar samples, providing evidence of the strong adhesion between the mortar and the fabric, as shown in Figure 9.

Finally, microscopic and interferometry analyses were carried out to gain deeper insights into the surface characteristics and irregularities of the specimen concentration and thickness of the residual fibers. The outcomes clearly highlight that samples with a water-to-cement ratio of 0.45 exhibited numerous surface irregularities. These irregularities led to a reduction in the interaction between the sheet and prisms, resulting in thinner layers of microfilaments adhering to their surfaces, as presented in Figure 10a. Conversely, for higher w/c ratios, the layer thickness of the remaining fibers substantially increased after the bond test, as demonstrated in Figure 10b,c. In addition, Table 6 shows the layer thickness of the residual microfilaments for the mixtures containing various w/c ratios.

### 3.2. Effect of Superplasticizer Content

In this phase of the experiment, the w/c ratio was lowered by incorporating the superplasticizer. In order to analyze the effect of SP content, 1.11% of SP by weight of cement was incorporated into the mixtures having a w/c ratio of 0.45 and 2.17% into the mixtures containing a w/c of 0.40. Their results were compared with the standard mortar that had a w/c ratio of 0.50 and 0.0% of SP. Figure 11 clearly illustrates a noteworthy reduction in the peeling force with the increase of superplasticizer content. Over a 28-day curing period, the peeling force decreased by 20.62% as the superplasticizer percentage increased from 0% to 1.11%. Furthermore, the force exhibited a more pronounced decline of 38.29% as the superplasticizer content was further raised from 1.11% to 2.17%.

In this part, minor variations in the surface textures of the samples were recorded despite maintaining constant workability. However, the specimens containing a lower w/c ratio and a greater quantity of superplasticizer exhibited larger pores and cavities; this is due to the lower quantity of cement paste on the surfaces of prisms and higher water bleeding, resulting in the reduction of the density of the remaining microfilaments, as shown in Figure 12a. In addition, smaller pores and much smoother surfaces were observed for the prisms with higher w/c ratios and a lower amount of superplasticizer because of a sufficient amount of cement paste and almost zero water bleeding, resulting in denser and thicker layers of remaining microfilaments, as presented in Figure 12b. It could be concluded that the surface texture, volume of cement paste, and water bleeding due to the superplasticizer had the largest effect on the adhesion between sheet and cement mortar.

To conclude, the microscopic and interferometry analyses provide a clear indication that specimens with a lower w/c ratio and a higher quantity of superplasticizer exhibited thinner layers of residual microfilaments on their surfaces. Conversely, the layer thickness of the remaining microfilaments was enhanced with a reduction in superplasticizer content and an increase in the w/c ratio, as illustrated in Figure 13. In addition, Table 7 shows the layer thickness of the residual microfilaments for the mixtures containing various percentages of SP.

Therefore, in order to have a strong adhesion between non-woven sheets and cementitious materials, it is recommended to have a mortar specimen with a smoother surface, fewer pores and cavities, and mixtures with a decreased amount of superplasticizer and an increased w/c ratio. However, enhancing the w/c ratio also has a limit because, beyond the limit, water bleeding could occur in the matrix, which will definitely decrease the adhesion between the tissue and mortar.

### 3.3. Mechanical Strengths

Furthermore, the mechanical tests were conducted on the same prisms after the bond test to confirm the intersection and adhesion of the residual microfilaments within the mortar mixtures. Overall, samples with non-woven sheets displayed higher mechanical strengths than the ones without such sheets (reference), independent of their mixture compositions. Notably, the prisms with a higher w/c ratio displayed an increase in flexural strength when compared to the ones with a lower w/c ratio, as shown in Figure 14a. Additionally, a marginal enhancement in the compressive strength was observed for the samples having higher water content compared to the lower ones, as shown in Figure 14b. This improvement can be attributed to the presence of the remaining fibers on the prism surfaces, which contribute to restraining early damage and augmenting load-bearing capacity. Moreover, the improvements in the mechanical strengths were more significant for the prisms with higher w/c ratios than those with lower ones. This distinction can be attributed to the thicker layer of residual microfilaments present on the surfaces of prisms with higher w/c ratios.

The experiments were extended to the prisms with varying percentages of superplasticizer. The results revealed that both compressive and flexural strengths were increased with the decrease of the SP content. Notably, these enhancements were more pronounced in the prisms with higher water content and a lower quantity of superplasticizer than those with a higher superplasticizer concentration. This difference could be attributed to the presence of a thicker layer of remaining microfilaments on specimens containing a higher w/c ratio and a reduced quantity of superplasticizer, as demonstrated in Figure 15.

## 4. Conclusions

The objective of this article was to examine how the w/c ratio and the percentage of superplasticizers impact the adhesion properties between non-woven fabrics and cement mortar. Based on the results obtained from the aforementioned experiment, the following key conclusions were drawn:In general, the adhesion between non-woven sheets and mortar specimens was consistently good using any w/c ratio or the quantity of superplasticizer;The peeling force notably increased as the w/c ratio was raised. This enhancement could be attributed to the smoother surface and sufficient presence of cement paste on the surfaces of prisms. In essence, specimens with a higher w/c ratio exhibited a greater density of microfilaments with thicker layers adhering to their surfaces, in contrast to those with lower w/c ratios. The thicker layer of the remaining microfilaments led to a larger contact area between the sheets and prisms, ultimately resulting in a stronger adhesion;Furthermore, augmenting the content of the superplasticizer exhibited a significant impact on bonding characteristics. The peeling force exhibited a notable decrease as water content was reduced and the superplasticizer amount was increased. This reduction can be attributed to the phenomenon of bleeding that occurs when a substantial quantity of superplasticizer is present alongside a reduced amount of cement paste on the prism surfaces. Additionally, the quantity and thickness of the remaining fibers reduced with the increase of superplasticizer content, leading to a corresponding decline in the peeling force;This phenomenon was clearly confirmed by interferometry and microscopic analyses. The observations revealed that reducing the w/c ratio or introducing a superplasticizer led to increased surface irregularities and porosity, coupled with a decrease in both the quantity and thickness of the remaining microfilaments;Finally, the mechanical properties of prisms were studied after the bond test. It was found that the flexural and compressive strengths have improved for the prisms, which have a higher w/c ratio and less superplasticizer.

## Figures and Tables

**Figure 1 polymers-15-04286-f001:**
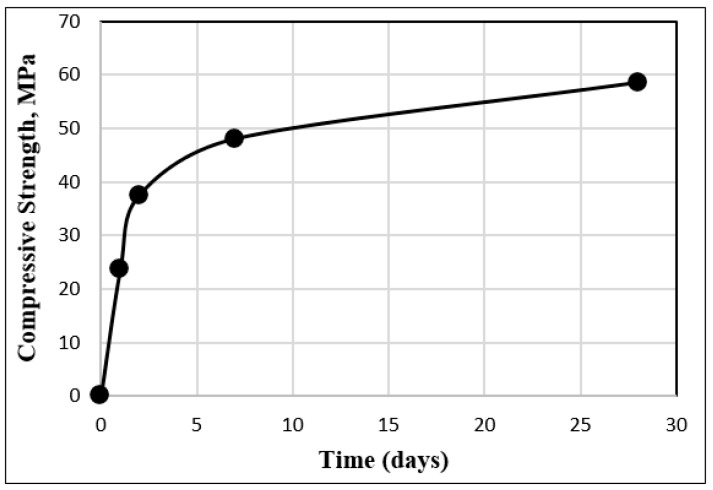
Increase in the compressive strength over time for a standard cement mortar using CEM I 52.5 N.

**Figure 2 polymers-15-04286-f002:**
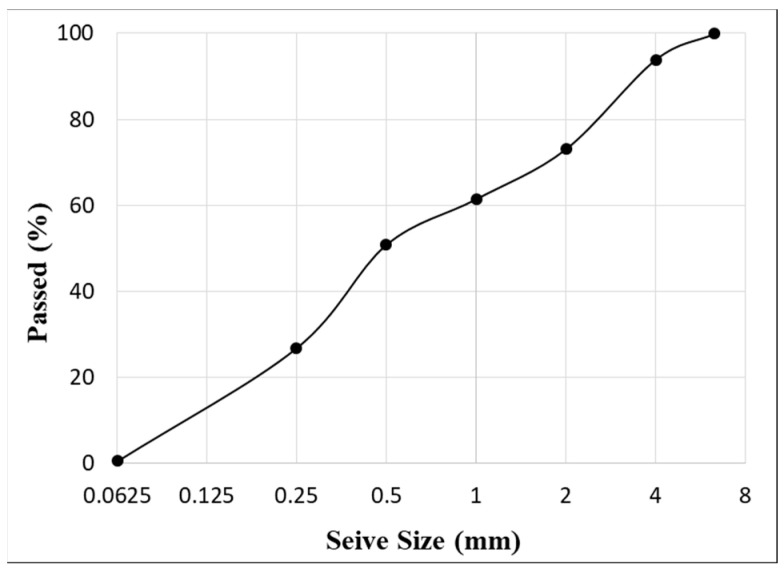
Size distribution of the sand (passing percentage through a set of sieves ranging from 0 to 6.3 mm).

**Figure 3 polymers-15-04286-f003:**
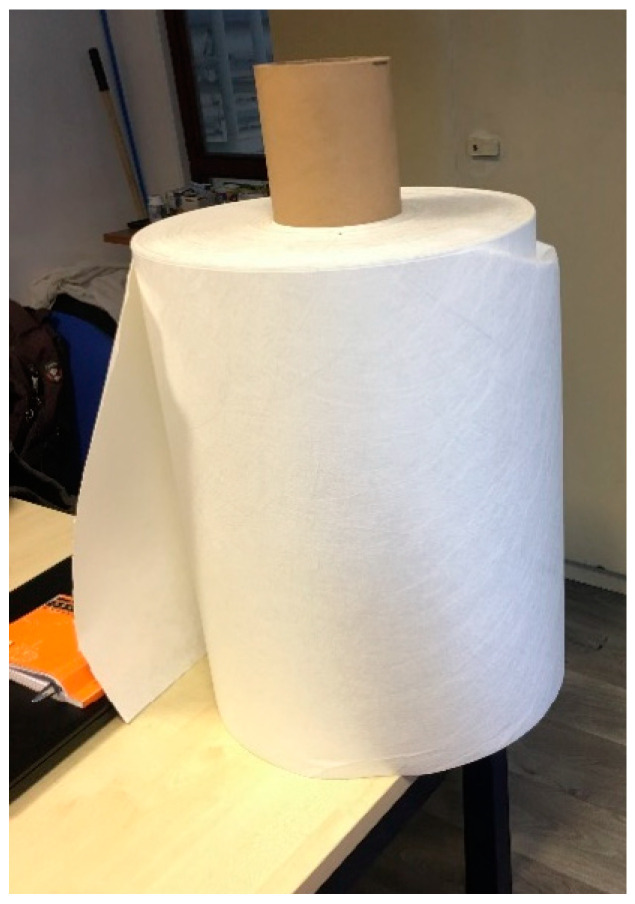
Non-woven plastic tissue.

**Figure 4 polymers-15-04286-f004:**
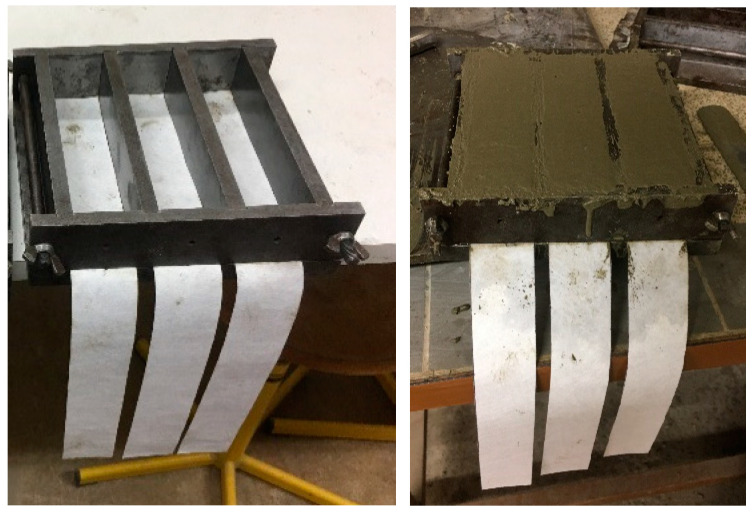
Placement of non-woven sheets in the molds.

**Figure 5 polymers-15-04286-f005:**
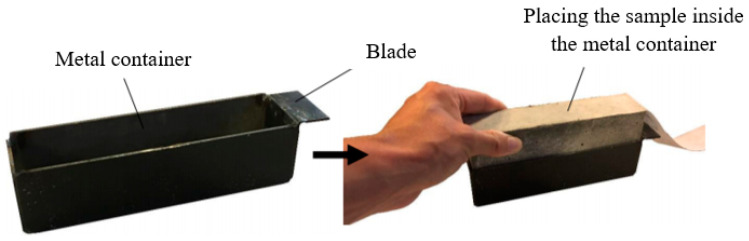
Metal container.

**Figure 6 polymers-15-04286-f006:**
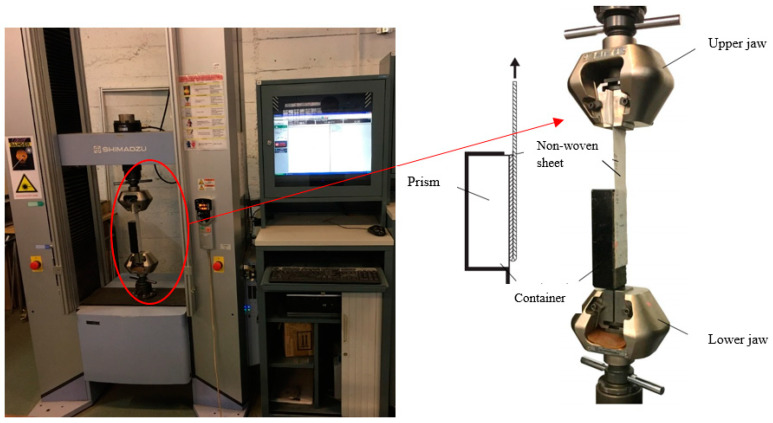
Shimadzu 100 kN machine and testing setup.

**Figure 7 polymers-15-04286-f007:**
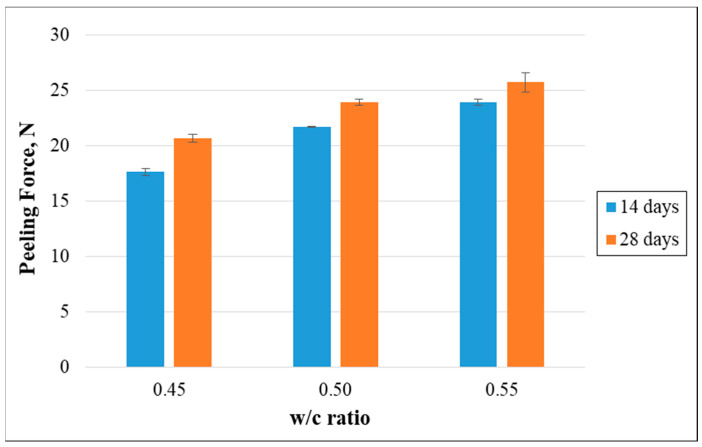
Effect of the w/c ratio on the peeling force.

**Figure 8 polymers-15-04286-f008:**
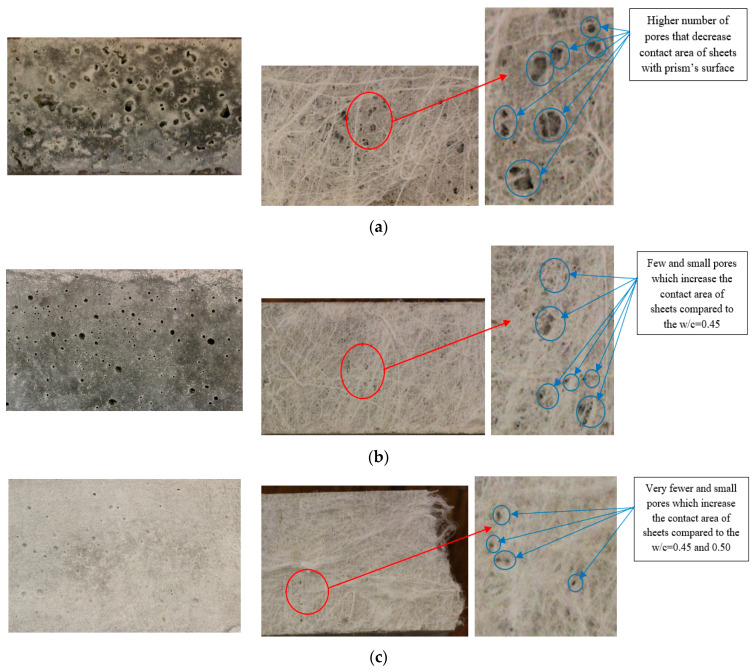
Surface texture and density of the remaining microfilaments on the surface of prisms: (**a**) w/c = 0.45, (**b**) w/c = 0.50, and (**c**) w/c = 0.55.

**Figure 9 polymers-15-04286-f009:**
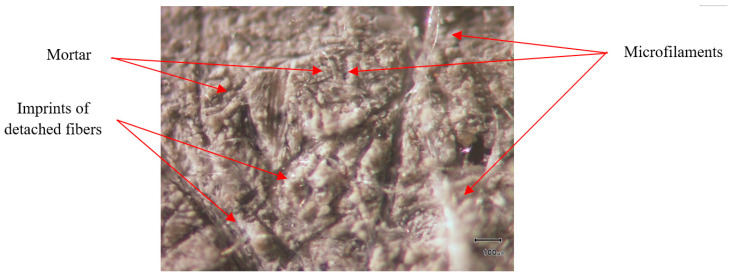
The closer view of the specimen under the microscope.

**Figure 10 polymers-15-04286-f010:**
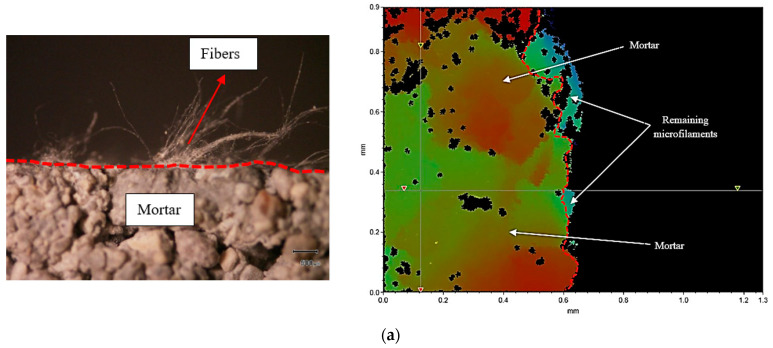
Surface irregularities and thickness of the remaining microfilaments on the surfaces of prisms: (**a**) w/c = 0.45, (**b**) w/c = 0.50, and (**c**) w/c = 0.55.

**Figure 11 polymers-15-04286-f011:**
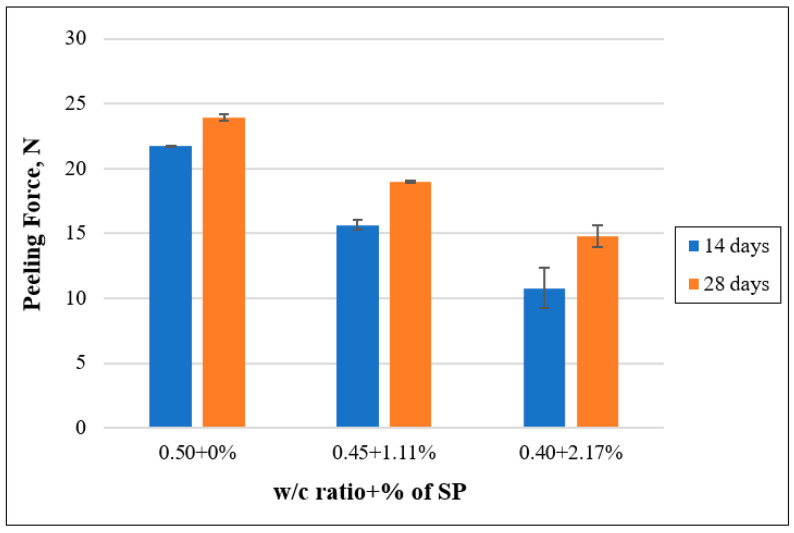
Effect of superplasticizer content on the peeling force.

**Figure 12 polymers-15-04286-f012:**
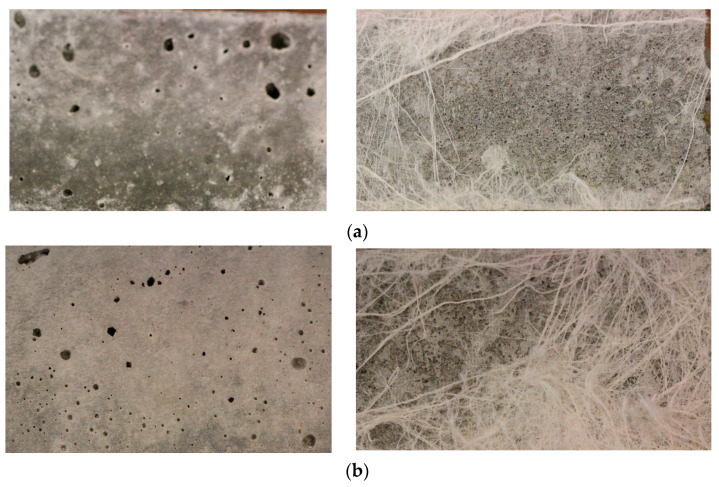
Surface texture and density of the remaining microfilaments on the surfaces of prisms: (**a**) w/c = 0.40 + 2.17% of SP, (**b**) w/c = 0.45 + 1.11% of SP, and (**c**) w/c = 0.50 + 0.0% of SP.

**Figure 13 polymers-15-04286-f013:**
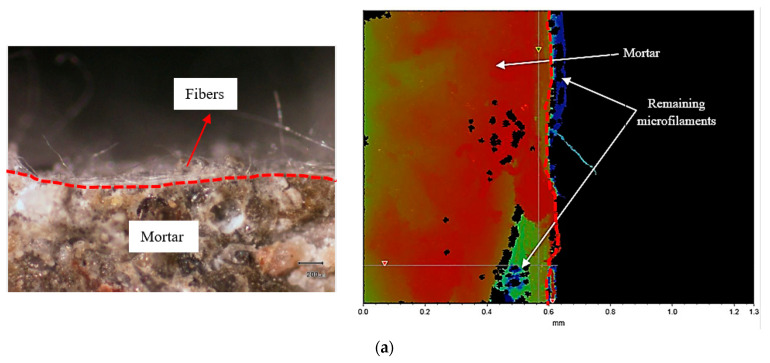
Microscopic and interferometry analyses of the specimens containing various percentages of SP: (**a**) w/c = 0.40, (**b**) w/c = 0.45, and (**c**) w/c = 0.50.

**Figure 14 polymers-15-04286-f014:**
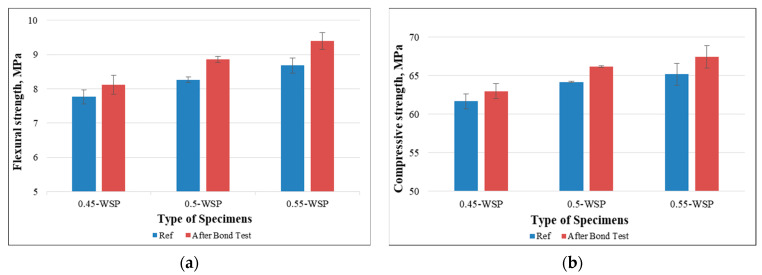
Mechanical strengths of samples after the bond test: (**a**) flexural strength and (**b**) compressive strength (effect of w/c ratio).

**Figure 15 polymers-15-04286-f015:**
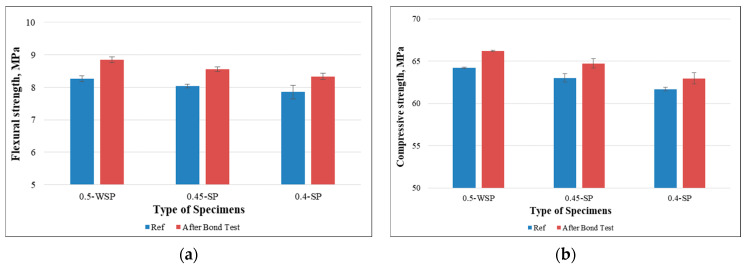
Mechanical strengths of samples after the bond test: (**a**) flexural strength and (**b**) compressive strength (effect of percentage of SP).

**Table 1 polymers-15-04286-t001:** Chemical composition of CEM I 52.5 N.

Components	Weight (%)	Phases	Weight (%)
CaO	61.3	C_3_S	63
SiO_2_	20		
Al_2_O_3_	4.8		
Fe_2_O_3_	3.1		
K_2_O	1.12	C_2_S	20
MgO	4.9		
Na_2_O	0.26		
S--	0.03		
Cl-	0.07	C_3_A	7
CO_2_	0.7		
SO_3_	3.7		
PAF	0.8		
INS	0.2		
CaO _Free_	1.6	C_4_AF	10
Na_2_O_eq active_	1		

**Table 2 polymers-15-04286-t002:** Chemical and physical properties of the superplasticizer [34].

Properties	Value/Type
Color	Dark brown
State	Liquid
Density	1150 ± 0.03 kg/m^3^
pH	7.5 ± 1.0
Chloride content	≤0.1%
Recommended dosage	5 dm^3^ for 1 m^3^ of concrete

**Table 3 polymers-15-04286-t003:** Physical and mechanical properties of non-woven sheets [26].

Properties	Unit	Test Method	Values
Web bonding	-	NF EN 29092 [39]	Hydrolase
Mas per unit area	g/m^2^	NF EN ISO 9073-1 [40]	Target 100
Thickness	mm	NF EN ISO 9073-2 [41]	Target 0.37
Tensile strength (machine direction)	N	NF EN ISO 13934-1 [42]	Target 300
Tensile strength (cross direction)	N	NF EN ISO 13934-1 [42]	Target 290
Tear strength (machine direction)	N	NF EN ISO 13937-1 [43]	Target 8
Tear strength (cross direction)	N	NF EN ISO 13937-1 [43]	Target 8
Elongation at break (machine direction)	%	NF EN ISO 13937-1 [43]	Target 45
Elongation at break (cross direction)	%	NF EN ISO 13937-1 [43]	Target 50
Water absorption	mL/m^2^	DIN 53923-78 [44]	Target 430 (Washed product)

**Table 4 polymers-15-04286-t004:** Mix proportion of the standard mortar [45].

Materials	Weight of Materials (g)
Sand	1350
Cement	450
Water	225

**Table 5 polymers-15-04286-t005:** Types of mixtures with various w/c ratios and percentages of SP content.

Number of Parameters	Sample Name	Cement (g)	Water (g)	% of SP (by Weight of Cement)	w/c Ratio	Slump (mm)
1	0.55-WSP	450	247.5	0.0	0.55	33.5
2	0.50-WSP	450	225.0	0.0	0.50	17.0
3	0.45-WSP	450	202.5	0.0	0.45	5.3
4	0.45-SP	450	202.5	1.11	0.45	17.5
5	0.40-SP	450	180.0	2.17	0.40	18.5

**Table 6 polymers-15-04286-t006:** Layer thickness of the remaining microfilaments for the mixtures containing various w/c ratios.

Mixture	Layer Thickness of the Remaining Microfilaments (mm)
0.45-WSP	0.08
0.50-WSP	0.25
0.55-WSP	0.52

**Table 7 polymers-15-04286-t007:** The layer thickness of the remaining microfilaments for the mixtures has various percentages of SP.

Mixture	Layer Thickness of the Remaining Microfilaments (mm)
0.40 + 1.11% SP	0.05
0.45 + 2.17% SP	0.08
0.50 + 0% SP	0.25

## Data Availability

The data presented in this study are available on request from the corresponding author.

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
