# Peer review of "Peeling Force Required for the Detachment of Non-Woven Plastic Tissue from the Surface of Mortar Prisms"

_polymers, 2023, doi:10.3390/polym15214286_

Round 1

Reviewer 1 Report

Comments and Suggestions for Authors

Dear Authors;

Thank you very much for giving me the opportunity to contribute to the quality of the manuscript entitled “Bond Properties between Non-Woven Plastic Tissue and Cement Mortar”.

Please consider the following comments and suggestions:

Title: The title is too general. “Bond properties” is not appropriate. Only detachment property at 180° was evaluated.

Abstract: It lacks of main quantitative results.

Introduction: Please, use English language in passive voice.

Materials: The general description of non-woven materials should be transferred to the introduction section. It lacks the technical description of the non-woven material used in the present work. What it is? What is the length distribution of fibers? What is the fiber diameter? What is the surface density of fibers? What is the strength of individual fibers and the non-woven material? Etc.

Preparation of the specimens: it is not clear how many specimens were prepared. 12 specimens is not statistically significant to evaluate 3 variables: curing time, W/C ratio and SP effects.

Testing setup and procedure: Please explain why ASTM D 90-98 norm was used.

Results and discussions: Is the reported adhesion force an averaged instant adhesion forces?

It would be interesting to compare the pores before and after the pouring process. Did these pores exists before the pouring process? At which extend?

Figure 11 is not correct. Caption refers to the effect of SP. But, the figure refers to W/C ratio.

Mechanical strengths: Flexural strength mechanical test is not described in the experimental section.

Comments on the Quality of English Language

English should be improved. Please, use passive voice or simple past.

Author Response

The authors would like to present their gratitude for the valuable comments of the reviewer and sincerely appreciate the deeper look at the manuscript.

The modifications along the manuscript were highlighted over track changes in the revised version

Reviewer 2 Report

Comments and Suggestions for Authors

The manuscript is very interesting. The work analysed the bond properties between the cement mortar and non-woven plastic sheets. In addition, the paper focused on the water/binder ratio and the amount of superplasticizer added. To be published, the paper needs minor revisions.

Abstract

Force bond” correspond to the adherence strength?

Line 14: the word “perfect” should be replaced.

Introduction

The introduction is well structured. However, the authors only presented three researches about the theme. The authors should improve the bibliography citing, including recent and relevant research papers presented in international journals dealing with this topic. Whitin 29 references of the paper only 1 is from 3 years ago, whereas the others are from more than 5 years ago, which may not reflect the latest research progress and not attract reader’s interests. The methodology to achieve the aim of the work should be written in the last paragraph of the introduction, namely the water to binder ratio used and the amount of superplasticizer.

Line 40: all the abbreviations should be written out in full when appears for the first time in the text, e.g. FRP, RC beams, CFRP, SNSM, NSM-FRP.

Materials and methods

The non-woven fabric used was a waste? Why did the authors choose this material?

Line 101: replace the word “possessed”.

Line 138: table 3 – replace “gr” for “g”.

Results and discussion

The results are well described in the manuscript. The figures are very detaile.

Conclusions

Line 318: replace the “excellent”.

Comments on the Quality of English Language

Minor editing of English required

Author Response

(The authors gave the same response as above.)

Reviewer 3 Report

Comments and Suggestions for Authors

Manuscript ID: polymers-2525118 (Type: Article)

Title: Bond Properties between Non-Woven Plastic Tissue and Cement Mortar

To the Authors

The introduction refers to the topic of the experiment. However, this part is developed in a very general way. References contains 24 author's items, which are often cited in groups, after short information. This needs improvement. It is suggested that information on the state of knowledge on the topic be provided in a more reliable manner.

The description of the experiment methodology requires improvement. Fig. 1 and Fig. 2 require correction (see attached file). The nonwoven fabric used for surface reinforcement is not specified. Giving information that it is made of plastic is too general.

The introduction describes the strengthening of structural concrete, and the experiment was performed on 4x4x16 cm beams made of cement mortar. In my opinion, concrete and mortar are cement composites with a slightly different structure (concrete also contains coarse aggregate, which has a different structure compared to the cement matrix), which may affect the different nature of adhesion of the fibrous fabric.

Additionally, a higher w/c value provides water that the fabric can absorb. This water provides more beneficial care at the interface between the cement matrix and the fibrous fabric. I can't find any analysis on this topic.

The report states that increasing the amount of superplasticizer (SP) causes bleeding, but there is no analysis on the example of determining the consistency of the mortar mixture. This should be added if there is such a statement.

The addition of SP to the mortar mixture is not precisely specified. You need to add what percentage of SP was added relative to the weight of cement. The reader must make his own calculations based on the information in table 4.

The attached file additionally highlights places for improvement.

Author Response

(The authors gave the same response as above.)

Round 2

Reviewer 1 Report

Comments and Suggestions for Authors

Dear Authors;

Thank you very much of addressing all comments and suggestions.

I recommend to accept the manuscript in its actual form after correction of few English spelling mistakes:

Lines 16 and 18: Feeling force by Peeling force.

Line 260: undr and microscop by under and microscope

Sincerely yours

Comments on the Quality of English Language

correction of few English spelling mistakes:

Lines 16 and 18: Feeling force by Peeling force.

Line 260: undr and microscop by under and microscope

Author Response

The authors would like to present again their gratitude for the deeper look of the reviewer at the manuscript.

The modifications along the manuscript are highlighted over track changes in the revised version.

Reviewer 3 Report

Comments and Suggestions for Authors

To the Authors

Supplements and corrections to the text significantly improved the quality of the paper.

It is important that the authors of publications respond positively to critical comments from reviewers. These comments are intended to help the authors of the paper improve the quality of their work, and this is what happened.

Author Response

(The authors gave the same response as above.)
